# Analysis of *CYP2C19* genetic variants with ischaemic events in UK patients prescribed clopidogrel in primary care: a retrospective cohort study

Luke C Pilling [ORCID],[1] Deniz Türkmen [ORCID],[1] Hannah Fullalove,[1] Janice L Atkins [ORCID],[1] Joao Delgado [ORCID],[1] Chia-Ling Kuo,[2,3] George A Kuchel,[3] Luigi Ferrucci,[4] Jack Bowden,[5] Jane A H Masoli,[1] David Melzer[1]

For numbered affiliations see end of article.

**Correspondence to**
Dr Luke C Pilling;
l.pilling@exeter.ac.uk

## ABSTRACT

**Objective** To determine whether *CYP2C19* loss-of-function (LoF) alleles increase risk of ischaemic stroke and myocardial infarction (MI) in UK primary care patients prescribed clopidogrel.

**Design** Retrospective cohort analysis.

**Setting** Primary care practices in the UK from January 1999 to September 2017.

**Participants** 7483 European-ancestry adults from the UK Biobank study with genetic and linked primary care data, aged 36–79 years at time of first clopidogrel prescription.

**Interventions** Clopidogrel prescription in primary care, mean duration 2.6 years (range 2 months to 18 years).

**Main outcome measure** Hospital inpatient-diagnosed ischaemic stroke, MI or angina while treated with clopidogrel.

**Results** 28.7% of participants carried at least one C*YP2C19* LoF variant. LoF carriers had higher rates of incident ischaemic stroke while treated with clopidogrel compared with those without the variants (8 per 1000 person-years vs 5.2 per 1000 person-years; HR 1.53, 95% CIs 1.04 to 2.26, p=0.031). LoF carriers also had increased risk of MI (HR 1.14, 95% CI 1.04 to 1.26, p=0.008). In combined analysis LoF carriers had increased risk of any ischaemic event (stroke or MI) (HR 1.17, 95% CI 1.06 to 1.29, p=0.002). Adjustment for aspirin coprescription produced similar estimates. In lifetables using observed incidence rates, 22.5% (95% CI 14.4% to 34.0%) of *CYP2C19* LoF carriers on clopidogrel were projected to develop an ischaemic stroke by age 79 (oldest age in the study), compared with 15.4% (95% CI 11.4% to 20.5%) in non-carriers, that is, 7.1% excess stroke incidence in LoF carriers by age 79.

**Conclusions** A substantial proportion of the UK population carry genetic variants that reduce metabolism of clopidogrel to its active form. In family practice patients on clopidogrel, *CYP2C19* LoF variants are associated with substantially higher incidence of ischaemic events. Genotype-guided selection of antiplatelet medications may improve outcomes in patients carrying *CYP2C19* genetic variants.

**Strengths and limitations of this study**

► This study included a relatively large sample of 7483 adults prescribed clopidogrel, and an extended follow-up period of up to 18 years.
► The data are from a retrospective cohort in UK Biobank, which recruited a relatively healthy population at baseline.
► Data were from linked electronic medical records (for prescribing and outcome data) and did not rely on potentially biased patient self-reporting.
► Electronic record data on prescriptions in the community were analysed, but did not include hospital inpatient prescription data.
► The sample had a maximum age of 79 years, so data on 80 plus year olds was not included.

## INTRODUCTION

Platelet aggregation, the process by which platelets adhere to each other, has long been recognised as critical for haemostatic plug formation and thrombosis.[1] Antiplatelet therapy that reduces platelet aggregation has become a central part of treatment to reduce cardiovascular and cerebrovascular disease incidence in at-risk patient groups.[2] Clopidogrel is an irreversible antagonist for the platelet $P2Y_{12}$ ADP receptor, thereby inhibiting platelet function and reducing likelihood of thrombosis.[3] In 2018, it was the 39th most commonly prescribed drug in the USA (20 million prescriptions).[4] Clopidogrel is a prodrug that requires transformation to its bioactive form by cytochrome P-450 (CYP) enzymes, primarily CYP2C19.[3] Loss-of-function (LoF) genetic polymorphisms in *CYP2C19* impair clopidogrel metabolism, with carriers of any LoF variant reported to have a 32.4% reduction in plasma active metabolite levels compared with non-carriers.[3] Clopidogrel resistance (high on-treatment

platelet reactivity) is predominantly caused by *CYP2C19* LoF variants, with a recent study of clopidogrel on-treatment reactivity reporting that 71.7% of LoF carriers show clopidogrel resistance, compared with 32.1% of non-carriers.[5] *CYP2C19* LoF carriers therefore have decreased platelet inhibition with consequently increased risk of thrombosis.[6]

LoF variants in the *CYP2C19* gene are common in many populations, with the *2 allele ranging from 15% frequency in Europeans (with ~27% of the population carrying at least one copy) to >30% in Asian populations.[7] A recent 2017 meta-analysis of 15 studies (12 East Asian, 3 European ancestry predominantly from the USA) included 4762 stroke or transient ischaemic attack (TIA) patients treated with clopidogrel; patients carrying any *CYP2C19* LoF alleles had nearly double the rates of incident strokes (risk ratio, RR 1.92, 95% CIs 1.10 to 2.06, p=0.01) compared with non-carriers.[8] However, most studies thus far focused on specific hospitalised patient groups with short (up to 1-year) follow-ups. A 2011 systematic review concluded that a significant association with stent thrombosis was due to small study effect bias and replication diversity.[9] Although the US Food and Drug Administration added a Boxed Warning to the clopidogrel label regarding poor metabolisers due to *CYP2C19* variants[10] in 2010, there is continued debate about whether the magnitude of effect of these variants on clinical outcomes justifies *CYP2C19* genotype[6] guided prescription or use of alternative medications unaffected by these variants. In 2019, a consensus statement from experts in the field recommended genotyping as an optional tool for guiding treatment in certain scenarios.[11]

Given the widespread continuing use of clopidogrel, and the current scarcity of evidence on clinical outcomes due to the LoF variants in the primary care setting, we aimed to estimate the association between *CYP2C19* LoF alleles and incident hospital-diagnosed ischaemic stroke and myocardial infarction (MI) in UK Biobank participants on treatment with clopidogrel. We also aimed to estimate longer-term (>1-year) outcomes, as these have previously been understudied, and to estimate outcomes of non-*CYP2C19* variants reported to have similar effects.[12]

## METHODS
### UK Biobank cohort
The UK Biobank recruited 503 325 community-based volunteers aged 40–70 years who visited one of 22 assessment centres in England, Scotland or Wales between 2006 and 2010.[13] Data collected at the baseline assessment included extensive questionnaires on demographic, health and lifestyle information. Anthropometric measures were also taken, in addition to blood samples for future biochemical and genetic analysis. This research was conducted under UK Biobank application 14631 (PI: DM). UK Biobank volunteers tended to be healthier than the general population at baseline assessment,[14] however, this largely prospective analysis of linked primary care

data is less affected by sample response patterns than baseline parameters.

### Prescription data
General practice prescriptions data are currently available for 230 096 (45.7%) participants, with >57 million prescribing events recorded. UK Biobank obtained data from the three main primary care electronic clinical record data providers (England: TPP/Vision; Scotland and Wales: EMIS/Vision). The extract date differs between the supplier's computer systems, and we used the following censoring dates: June 2016 (England), April 2017 (Scotland), and September 2017 (Wales). Detailed description of the data extracted and limitations are available from UK Biobank.[15] Data include prescription dates, drug code (in clinical Read V.2, British National Formulary or Dictionary of Medicines and Devices format (depending on supplier), drug name and quantity, where available. We identified all prescribing events for clopidogrel and aspirin. Details for drug names and clinical codes used are available in online supplemental information and table 1.

In the prescribing data, we identified all records matching a clopidogrel code and for each patient determined the date first prescribed, date last prescribed, number of total prescriptions and average number of prescriptions over the time period. We analysed data for patients who received two or more prescriptions, with at least one clopidogrel prescription for each 2-month period between the dates first/last prescribed. This last criterion excluded participants with longer gaps between prescriptions, who may not have been taking clopidogrel for the entire study period, thus minimising immortal time bias.[16]

Participants never prescribed clopidogrel were used for sensitivity analysis of *CYP2C19* LoF variants and relevant vascular outcomes.

### *CYP2C19* genetic variants
Directly genotyped genetic variants (n=805 426) were available for 488 377 UK Biobank participants, based on two almost identical genotyping platforms sharing >95% of variants: the Affymetrix Axiom UKB array (in 438 427 participants) and the Affymetrix UKBiLEVE array (in 49 950 participants). After extensive quality control by the central UK Biobank team,[17] genotype imputation was successful in 487 442 participants and increased the number of genetic variants to ~96 million.[17] Our primary analysis included 451 367 participants (93%) identified as genetically European (identified by genetic clustering, as described previously[18]), to reduce bias in genetic studies from population stratification, which arises when different ancestral groups are analysed together in genetic analyses.[19] Secondary analysis in other ancestral groups was not possible due to low numbers in the clopidogrel patients (n=492 non-Europeans with sufficient clopidogrel data; this group included all other ancestries such as South or East Asian, African and mixed).

We analysed the *CYP2C19* genetic variants (*2-*8 and *17) with well-documented effects on clopidogrel metabolism in the literature[9] and in the PharmGKB database[12]: three loss-of-function alleles (*2, *3 and *8; intermediate or poor metabolisers) and one gain-of-function allele (*17; rapid metaboliser) were either directly genotyped (*3 and *8) or imputed with high confidence (*2 and *17:>99.9% imputation confidence). *CYP2C19*4 was imputed with <80% imputation confidence (74.4%) so was excluded from analysis (*4 is very rare so this has minimal impact on the analysis: with 0.25% allele frequency reported in the GnomAD[7] database https://gnomad.broadinstitute.org/). *CYP2C19* *5, *6 and *7 were not available in the genotyping data, as they are extremely rare, especially in European populations (<0.1% allele frequency in GnomAD).[7] See online supplemental table 2 for details on individual variants used.

### Non-*CYP2C19* genetic variants

We also investigated non-*CYP2C19* genetic variants that affect clopidogrel from two sources: we searched the PharmGKB database[12] for variants classified as being supported by moderate or high clinical annotation levels of evidence, that is, associations that replicated in subsequent studies, even if the original study was small. In addition to variants in *CYP2C19* (level 1 'high' levels of evidence) one variant in *CES1* has level 2B 'moderate' evidence: rs71647871 was directly genotyped on the microarray but is rare, with only 249 heterozygotes in the clopidogrel analysis group (no homozygotes).

Lastly, we investigated variants identified in a genome-wide association study of clopidogrel active metabolite levels (rs187941554 and rs80343429).[20] Both variants are available in the UK Biobank imputed genotype data: rs187941554 was imputed with high confidence (99.9%), however, the overall rs80343429 imputation accuracy was 81.7% (n=3 individual participants with low-confidence genotype calls—imputed genotype dose >0.25 and<0.75 were excluded from analysis).

### Disease ascertainment

Diagnosis of ischaemic stroke and MI ascertainment from hospital admissions records (Hospital Episode Statistics, HES) were available up to 14 years follow-up after baseline assessment, covering the entire period up to the date of censoring of primary care prescribing data (HES in England up to 30 September 2020: data from Scotland and Wales censored to 31 August 2020 and 28 February 2018, respectively). Diagnosis of ischaemic stroke was ascertained using International Classification of Diseases, Tenth Revision (ICD-10) code I63*. Diagnosis of MI was ascertained using ICD-10 codes I21*; I22*; I23*; I24*; I25*. In sensitivity analysis we investigated major bleeding using ICD-10 codes from[21] (see online supplemental table 3 for codes used).

For prevalent ascertainment of diagnoses at baseline of TIA, MI/angina, atrial fibrillation, hypertension or peripheral vascular disease we used primary care records.

Diagnosis codes (read V.2) were downloaded from the CALIBER phenotype library (https://portal.caliberresearch.org/disease-or-syndrome).[22] Corresponding CTV3 codes were identified using the UK Biobank 'Clinical coding classification systems and maps' resource (https://biobank.ctsu.ox.ac.uk/crystal/refer.cgi?id=592).

### Statistical analysis

Data are from multiple sources (demographic and genetic data from UK Biobank assessment, clopidogrel prescribing data from linked electronic primary care records, and stroke and MI outcomes from linked electronic hospital inpatient records). Patients with any missing data needed for analysis were excluded, with details shown in the results.

Estimates of associations between genotype and incident disease were obtained from Cox's proportions hazards regression models, with adjustment for age at first prescription, sex and the first 10 principal components of genetic ancestry (to control for population substructure). Patients entered the model 7 days after the date of first prescription of clopidogrel; this was to exclude the small number of events that occurred immediately after being prescribed clopidogrel, before it could reasonably have had a preventative effect. Seven days was chosen because the effect of clopidogrel on platelet aggregation was significantly different between *CYP2C19* genotype groups after 7–10 days in a study of 375 patients.[23] In sensitivity analysis we included all events during the prescribing period. Patients exited the model on the last known date of clopidogrel prescription. STATA (V.15.1) software was used for analysis. We visually inspected Kaplan-Meier plots and used STATA function 'estat phtest, detail' to generate Schoenfeld residuals to test for violations of the proportional-hazards assumption.

Population attributable fraction (PAF) in this analysis is the estimated proportion of all cases that would not have occurred if no participants carried the clopidogrel modifying genetic variants. It was calculated by multiplying the proportion of incident ischaemic strokes in the genotype carriers by (1–1/HR for ischaemic stroke) and similarly for MI.[24] This method is preferred to simple observed versus expected cases as it uses the estimate from the time-to-event model, which has been adjusted for covariates.[24] Lifetable probabilities of incident outcomes were estimated from age 37–79 (the minimum and maximum ages the patients were first prescribed clopidogrel, respectively) by *CYP2C19* genotype, applying observed incidence rates in each age group to a notional cohort (where all participants in the analysis had reached the maximum age group), estimating cumulative incident case numbers, using STATA command 'sts list'.

### Patient and public involvement

Patients and participants were and are extensively involved in the UK Biobank study itself. No patients were involved in developing the research question or the outcomes tested. A lay abstract was reviewed by members

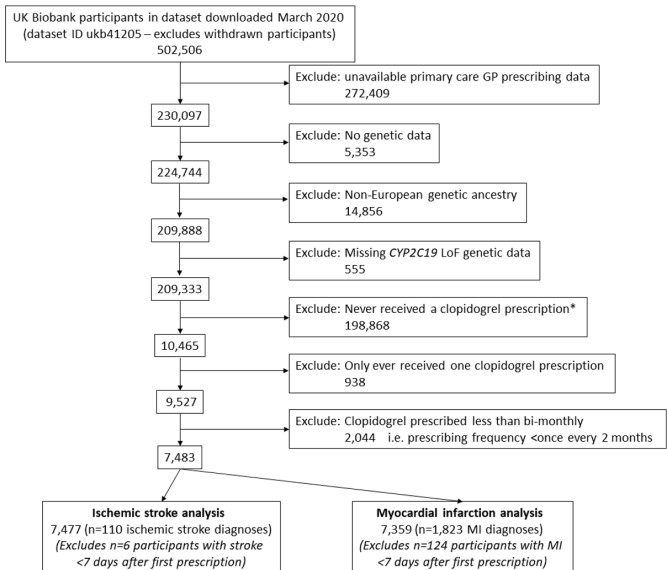

**Figure 1** UK Biobank participants eligible for analysis: cohort flow chart. Cohort flow chart shows how the eligible participants to study were identified. Summary statistics for the 7483 participants are available in table 1. *Participants never prescribed clopidogrel were used for sensitivity analysis of CYP2C19 LoF variants and vascular outcomes. GP, general practitioner; LoF, loss-of-function; MI, myocardial infarction.

of the Peninsula Public Engagement Group in Exeter, who provided positive feedback on the importance of the study from a patient perspective.

## RESULTS

A total of 7483 UK Biobank participants of European ancestry met inclusion criteria, with at least two prescriptions of clopidogrel recorded in primary care data (see figure 1 for cohort flow chart with detailed inclusion/exclusion criteria). The mean age at first prescription was 64.3 years (SD 7.2) and 2563 were female (34.2%). The clopidogrel exposure period ranged from 2 months to 18 years (mean 2.59 years, SD 2.98). Cohort selection is described in table 1. Patients carrying at least one *CYP2C19* LoF variant (*2-*8, intermediate/poor clopidogrel metabolisers) did not have significantly different likelihood of receiving clopidogrel compared with non-carriers (OR 1.01: 95% CIs 0.96 to 1.05, p=0.75) and had similar prevalence of cardiovascular co-morbidities prior to diagnosis (table 1).

### *CYP2C19* LoF associations with incident ischaemic stroke

Incident hospital-diagnosed ischaemic stroke (cerebral infarction) was identified in 110 of 7477 participants (1.47%) eligible for analysis (after excluding n=6 who had stroke within 7 days of first prescription). *CYP2C19* LoF (*2-*8, intermediate/poor metabolisers) carriers (n=2144, 28.7%) were more likely to have an ischaemic stroke while on treatment for clopidogrel than patients without *CYP2C19* LoF genetic variants (HR 1.53, 95% CI

1.04 to 2.26, p=0.031). See table 2 for details and figure 2A for Kaplan-Meier plot. 1.29% of the patients without any *CYP2C19* LoF genetic variants had an stroke during the prescribing period (mean 2.59 years between first and last clopidogrel prescription, SD 2.98) compared with 1.91% of the *CYP2C19* LoF carriers: this absolute 0.64% excess is statistically significant (95% CI 0.021% to 1.35%, p=0.027).

To test whether the effect of *CYP2C19* LoF genotypes on ischaemic stroke risk was specific to clopidogrel, and not via a separate but unknown biological mechanism, we analysed the 198868 UK Biobank participants with general practitioner (GP) prescribing data in whom clopidogrel was never prescribed (see figure 1). There were 512 ischaemic stroke events in this group after baseline assessment and before the end of Feb 2016 (earliest censoring date for GP prescribing data). *CYP2C19* LoF carriers were not at significantly increased risk of ischaemic strokes compared with non-carriers (HR 1.12, 95% CI 0.93 to 1.35, p=0.24).

To calculate the PAF, we multiplied the proportion of incident ischaemic strokes that occurred in that patients who had intermediate or low metaboliser variants (41/110) by (1–1/HR for ischaemic stroke=1–1/1.53=0.346) giving 0.129 (see the Methods section). We, therefore, estimate the PAF for ischaemic stroke—the proportion of ischaemic strokes in this population that would not have occurred if clopidogrel efficacy was not affected by *CYP2C19* LoF variants—to be 12.9% (95% CI 1.4% to 20.8%). Kaplan-Meier curves for diagnosis of ischaemic strokes in participants prescribed clopidogrel are shown in figure 2. In lifetable estimates based on observed incidence rates from 36 to 79 (the minimum and maximum ages at which clopidogrel was first prescribed) the risk of ischaemic stroke in CYP2C19 LoF carriers was 22.5% (95% CI 14.4% to 34.0%), compared with 15.4% (95% CI 11.4% to 20.5%) in non-carriers.

We performed secondary analyses estimating outcomes for participants prescribed clopidogrel for >1year (n=4316: 57.8%). *CYP2C19* LoF carriers had significantly increased risk of strokes (HR 1.63, 95% CI 1.01 to 2.64, p=0.047) in this period (table 2). However, the association with only those events within the first year of prescription was not statistically significant (HR 1.33, 95% CI 0.69 to 2.57, p=0.390). In tests of the proportional-hazards assumption (see the Methods section) in the model using all available events there was no evidence that the HR changed over time (p=0.42). The analysis using all participants suggests LoF carriers are at increased risk for the duration of the clopidogrel prescribing (HR 1.53, 95% CI 1.04 to 2.26, p=0.031).

In sensitivity analysis we adjusted the main analysis of *CYP2C19* LoF carriers for history of stroke, TIA, MI/angina, PAD, and atrial fibrillation to assess whether disease indication modified the effect of genotype on ischaemic stroke risk and found the estimate to be unaffected (HR 1.56, 95% CI 1.06 to 2.31, p=0.024). We also performed analysis including all events during the

**Table 1** Summary of UK Biobank participants with GP-prescribed clopidogrel

| | CYP2C19 genotype | |
|---|---|---|
| | Normal metaboliser (*1/*1) | Intermediate/poor (any *2-*8) |
| No of participants (% of total n=7483) | 5338 (71.3) | 2145 (28.7) |
| Females, n (% of genotype group) | 1847 (34.6) | 712 (33.2) |
| Age at first clopidogrel prescription | | |
| Minimum : maximum | 36.5 to 78.8 | 37.6 to 78.1 |
| Mean (SD) | 64.1 (7.3) | 64.3 (7.2) |
| Diagnoses* prior to clopidogrel prescription, n (%) | | |
| Ischaemic stroke | 509 (9.5) | 240 (11.2) |
| Transient ischaemic attack | 1142 (21.4) | 422 (19.7) |
| Myocardial infarction/angina | 2980 (55.8) | 1267 (59.1) |
| Atrial fibrillation | 408 (7.6) | 180 (8.4) |
| Hypertension | 3072 (57.6) | 1277 (59.5) |
| Peripheral vascular disease | 442 (8.3) | 177 (8.3) |
| Any of the above six diagnoses | 4863 (91.1) | 1982 (92.4) |
| No of clopidogrel prescriptions | | |
| Minimum : maximum | 2 to 467.0 | 2 to 494.0 |
| Mean (SD) | 28.3 (36.7) | 27.9 (36.9) |
| Years between first and last clopidogrel prescription | | |
| Minimum : maximum | 0.2 to 17.9 | 0.2 to 17.3 |
| Mean (SD) | 2.6 (3.0) | 2.5 (2.9) |
| Also prescribed aspirin while taking clopidogrel, n (%) | 2632 (49.3) | 1119 (52.2) |
| Ischaemic stroke while taking clopidogrel, n (%) | 69 (1.3) | 41 (1.9) |
| Myocardial infarction while taking clopidogrel, n (%) | 1268 (24.1) | 554 (26.4) |

Analysis of European-ancestry participants with >1 clopidogrel prescription in the available GP prescribing data. Participants excluded if clopidogrel prescribing frequency was less than once every 2 months. Events occurring after the last known date of clopidogrel prescription are also excluded.
*Ischaemic stroke diagnoses are from hospital in-patient records only; all other disease also include diagnoses recorded in primary care.
GP, general practitioner.

prescribing period (the primary analysis excluded events in the first 7 days of prescribing because the effect of clopidogrel on platelet aggregation was significantly different between CYP2C19 genotype groups after 7–10 days in a study of 375 patients).[23] This period included six strokes and changed the estimates from (HR 1.53, 95% CI 1.04 to 2.26, p=0.031) to (HR 1.45, 95% CI 0.99 to 2.13, p=0.053). Though the 95% CIs cross the null the association is highly consistent, despite inclusion of some possibly invalid events.

We investigated haemorrhagic strokes (ICD-10 codes I61* and I62*), however, there were too few diagnoses made (n=9 in the 7483 participants included in the study) during the clopidogrel prescribing period, so no analysis was performed.

We also analysed outcomes for the *17 (gain of function) genotype carriers (rapid metabolisers). *17 carriers were not at altered risk of ischaemic stroke during the prescribing period (HR 0.98, 95% CI 0.61 to 1.57, p=0.930, online supplemental table 4) compared with *1/*1 patients. In analysis of Major Bleeding while prescribed clopidogrel CYP2C19 *17 carriers were not at significantly greater or lesser risk compared with non-carriers (n events during prescribing period=167; OR 0.88, 95% CI 0.62 to 1.24, p=0.46).

### CYP2C19 LoF associations with incident myocardial infarction

Incident hospital-diagnosed MI occurred in 1823 of 7359 participants (24.8%), after excluding 124 subjects who had MIs within 7 days of first clopidogrel prescription. CYP2C19 LoF (*2-*8, intermediate/poor metabolisers) carriers (n=2100, 28.5%) were more likely to have an MI while being prescribed clopidogrel than patients without CYP2C19 LoF genetic variants (HR 1.14, 95% CI 1.04 to 1.26, p=0.008, see table 2 for details and figure 2B for Kaplan-Meier plot). The PAF for MI was 3.73% (95% CI 1.2% to 6.3%).

We performed secondary analysis investigating the participants prescribed clopidogrel for >1 year (n=3511: 47.7%). CYP2C19 LoF carriers were not at significantly

**Table 2** *CYP2C19* genotype associations with incident ischaemic stroke and MI events in patients prescribed clopidogrel

| Outcome, model | *CYP2C19 genotype* | N | N cases | Person-years | HR | 95% CIs | P value |
|---|---|---|---|---|---|---|---|
| **Ischaemic stroke** | | | | | | | |
| All participants | Normal metaboliser (*1/*1) | 5333 | 69 | 13 248 | | | |
| | Intermediate/poor (any *2-*8) | 2144 | 41 | 5110 | 1.53 | 1.04 to 2.26 | 0.031 |
| | Total | 7477 | 110 | 18 358 | | | |
| Events in first year only | Normal metaboliser (*1/*1) | 5333 | 25 | 4419 | | | |
| | Intermediate/poor (any *2-*8) | 2144 | 14 | 1789 | 1.33 | 0.69 to 2.57 | 0.390 |
| | Total | 7477 | 39 | 6208 | | | |
| Events after first year only | Normal metaboliser (*1/*1) | 3071 | 44 | 8829 | | | |
| | Intermediate/poor (any *2-*8) | 1245 | 27 | 3322 | 1.63 | 1.01 to 2.64 | 0.047 |
| | Total | 4316 | 71 | 12 151 | | | |
| **MI** | | | | | | | |
| All participants | Normal metaboliser (*1/*1) | 5259 | 1269 | 10 093 | | | |
| | Intermediate/poor (any *2-*8) | 2100 | 554 | 3709 | 1.14 | 1.04 to 1.26 | 0.008 |
| | Total | 7359 | 1823 | 13 803 | | | |
| Events in first year only | Normal metaboliser (*1/*1) | 5255 | 906 | 3831 | | | |
| | Intermediate/poor (any *2-*8) | 2099 | 415 | 1511 | 1.16 | 1.03 to 1.31 | 0.011 |
| | Total | 7354 | 1321 | 5342 | | | |
| Events after first year only | Normal metaboliser (*1/*1) | 2529 | 362 | 6258 | | | |
| | Intermediate/poor (any *2-*8) | 982 | 139 | 2198 | 1.09 | 0.90 to 1.33 | 0.370 |
| | Total | 3511 | 501 | 8455 | | | |

Analysis of European-ancestry participants with >1 clopidogrel prescription in the available GP prescribing data. Participants excluded if clopidogrel prescribing frequency was less than once every 2 months. Events <1 week after first clopidogrel prescription are excluded. Events occurring after the last known date of clopidogrel prescription are also excluded. HRs from Cox's proportional hazards regression models adjusted for age at first clopidogrel prescription, sex and genetic principal components of ancestry 1–10.
GP, general practitioner; MI, myocardial infarction.

increased risk of MIs (HR 1.09, 95% CI 0.90 to 1.33, p=0.37) in this period (table 2). However, LoF carriers did have significantly increased risk of MI within the first year of clopidogrel prescription (HR 1.16, 95% CI 1.03 to 1.31, p=0.011). In tests of the proportional-hazards assumption (see the Methods section) in the model using all available events there was no evidence that the HR changed over time (p=0.70). The analysis using all participants suggests LoF carriers are at increased risk for the duration of the clopidogrel prescribing (HR 1.14, 95% CI 1.04 to 1.26, p=0.008).

We also analysed the *17 (gain of function) genotype carriers (rapid metabolisers). *17 carriers did not have significantly different risk of MI during the prescribing period (HR 0.95, 95% CI 0.85 to 1.06, p=0.33, online supplemental table 4) compared with *1/*1 patients.

The association between *CYP2C19* LoF carriers and MI in participants never prescribed clopidogrel (n=4390 MIs) was not significant (HR 1.03, 95% CI 0.96 to 1.10, p=0.41). Sensitivity analysis including all MIs during the prescribing period (ie, not excluding those within 7 days of first known prescription) included 126 more events and changed the estimates from (HR 1.14, 95% CI 1.04 to 1.26, p=0.008) to (HR 1.16, 95% CI 1.06 to 1.28, p=0.021).

In secondary analysis of any ischaemic event (stroke or MI, n=1886 events in 7354 patients included in analysis)

*CYP2C19* LoF (*2-*8, intermediate/poor metabolisers) carriers were more likely to have an event while being prescribed clopidogrel than patients without *CYP2C19* LoF genetic variants (HR 1.17, 95% CIs 1.06 to 1.29, p=0.002).

### Dual antiplatelet therapy: clopidogrel coprescribing with aspirin
Of 7483 patients prescribed clopidogrel 3730 (49.9%) were also prescribed aspirin at least once during the analysis period. *CYP2C19* LoF carriers had slightly higher likelihood of ever receiving a prescription of aspirin compared with non-carriers during their clopidogrel exposure period (n=1118 (52.2%) vs n=2629 (49.3%), OR from logistic regression 1.14: 95% CI 1.02 to 1.26, p=0.017). Ever being prescribed aspirin did not significantly interact with *CYP2C19* LoF carrier status in the model of incident stroke or MI (p>0.05). The association between *CYP2C19* LoF carrier status and stroke or MI was consistent (although attenuated due reduced sample sizes in subgroups) in participants who received an aspirin prescription (n=3730. Stroke HR 1.61, 95% CI 0.87 to 2.98, p=0.13; MI HR 1.10, 95% CI 0.98 to 1.24, p=0.10) and those who did not (n=3753. Stroke HR 1.48, 95% CI 0.89 to 2.46, p=0.13; MI HR 1.19, 95% CI 0.98 to 1.45, p=0.087).

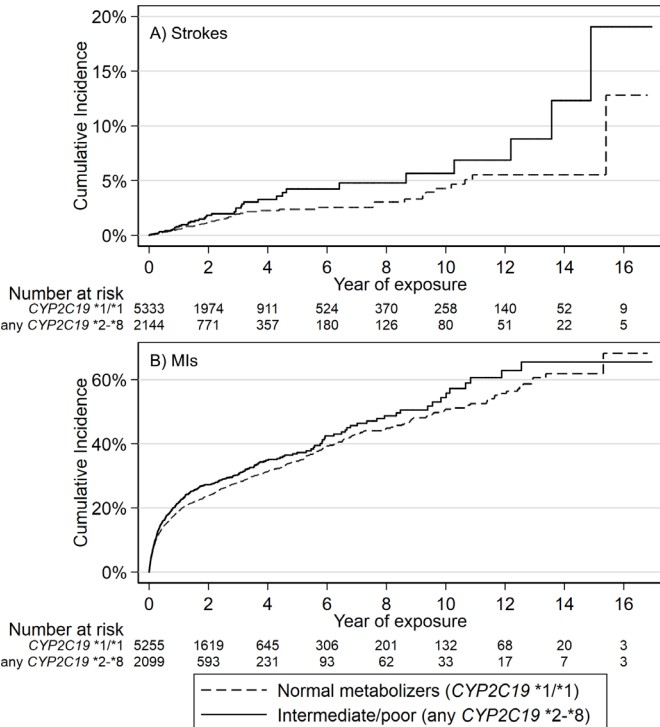

**Figure 2** Kaplan-Meier plots of strokes and MIs in patients prescribed clopidogrel, stratified by CYP2C19 loss of function genotypes. Kaplan-Meier failure plots for (A) incident ischaemic stroke and (B) incident MI showing cumulative hazard (%) with increasing time taking clopidogrel in patients prescribed clopidogrel for at least 2 months, stratified by CYP2C19 genotype (carriers of any *2-*8 loss of function variant, 'intermediate/poor metabolisers of clopidogrel,' vs *1/*1 normal metabolisers). MIs, myocardial infarctions.

### Analysis of non-*CYP2C19* genetic variation associated with clopidogrel active metabolite levels

A rare variant in *CES1* (rs71647871, n=249 CT heterozygotes) with moderate clinical annotation evidence for clopidogrel on the PharmGKB database[12] was not significantly associated with strokes or MIs (stroke HR 0.86, 95% CI 0.27 to 2.70, p=0.79; MI HR 1.26, 95% CI 0.99 to 1.60, p=0.057), although the MI association was borderline.

Two genetic variants identified in a GWAS of clopidogrel active metabolite levels[20] were also analysed. Despite its low frequency in the population, rs187941554 (n=32 GA heterozygotes, 0.4% of 7,468) was associated with substantially raised risk of incident stroke (n strokes=2; HR 5.44, 95% CI 1.33 to 22.30, p=0.019) but not MIs (HR 1.20, 95% CI 0.62 to 2.32, p=0.58). The rs80343429 A allele was not associated with increased risk of stroke or MI (stroke HR 1.19, 95% CI 0.68 to 2.09, p=0.53; MI HR 1.09, 95% CI 0.94 to 1.26, p=0.25).

### DISCUSSION

Clopidogrel requires metabolic activation to be effective, mainly by the CYP2C19 liver enzyme. *CYP2C19* LoF genetic variants (implying intermediate or low metaboliser

status) have been shown to increase stroke and MI rates mainly in hospitalised patients with acute ischaemic conditions (or undergoing related interventions such as cardiac stents[25]), with little data on outcomes in primary care or over longer follow-up periods. In this study of 7483 UK Biobank primary care patients prescribed clopidogrel, carriers of *CYP2C19* LoF alleles had substantially increased risks of incident ischaemic strokes and MI. We also showed that carriers of LoF alleles are at increased risk of stroke even during longer-term (>1-year) therapy; the effect is not limited to acute stroke patients.

Our results are consistent with previous studies of clopidogrel efficacy in *CYP2C19* LoF genotype carriers, although the previous evidence was predominantly from high-risk acutely hospitalised patient groups,[26–29] with limited follow-up periods (rarely over 12 months). Some studies reported that the effect of *CYP2C19* genotype on reducing clopidogrel efficacy was limited to the short term (1–6 months) only,[30 31] but in our community based longer term follow-up we found no evidence for deviation from the proportional hazards assumptions, with survival analysis results suggesting sustained effects over the follow-up period. We had limited sample size, and therefore, limited statistical power to estimate effects for specific periods, but the intermediate or low metaboliser variant association with excess strokes was significant in the subgroup prescribed clopidogrel for more than 1 year.

A recent 2019 meta-analysis of 16 randomised clinical trials (RCTs)[32] found that dual antiplatelet therapy with aspirin was associated with significantly lower stroke rates (RR 0.80, 95% CI 0.72 to 0.89), however, some evidence of increased major bleeding was also observed (RR 1.90, 95% CI 1.33 to 2.72). The authors noted that common *CYP2C19* LoF variants may account for the effectiveness of dual therapy over clopidogrel monotherapy.[32] In our study, we found LoF carriers still had increased stroke or MI risk, even if coprescribed aspirin.

Clopidogrel is a recommended antiplatelet medication for the prevention of stroke in the USA[2] and is the currently recommended antiplatelet medication for prevention of stroke or TIA in the UK.[33] However, antiplatelet medications such as ticagrelor and prasugrel are not dependent on CYP2C19 metabolism and reportedly have better clinical outcomes in LoF carriers compared with clopidogrel.[34] A recent meta-analysis of 16 studies (8 RCTs and 8 cohorts) concluded that genotype-guided antiplatelet treatment (primarily based on *CYP2C19* LoF alleles) improved patient outcomes for major cardiovascular events (eg, RR for MI 0.45, 95% CI 0.35 to 0.58, p<0.00001), with decreased risk of major bleeding.[29] Treatment strategy in the different metaboliser groups varied between individual studies, but included use of higher doses of clopidogrel (150 mg/day instead of the usual 75 mg/day), or prescribing of prasugrel or ticagrelor, or a combination. Other recent studies in different populations, especially in patients with acute coronary syndrome and those undergoing primary percutaneous coronary intervention such as the large POPular Genetics trial, also

support this conclusion.[26–28 35 36] Therefore, further work is needed in primary care populations to inform optimal antiplatelet care, given our estimate that over 12% of strokes in the UK study population on clopidogrel could be avoided if low and intermediate metabolisers could achieve the same clinical outcomes as are being achieved in the normal metaboliser group. Economic analysis in the UK comparing clopidogrel (out of patent at the time of writing) to alternatives not necessarily out of patent are needed.

This is the largest single study of primary care patients prescribed clopidogrel, incorporating *CYP2C29* genetic data and longer (>1 year) follow-up of clinical outcomes. However, this study is not without limitations. UK Biobank participants tended to be healthier than the general population at baseline assessment,[14] however, linked primary care data used follow-up analysis should be unaffected by sample response patterns. Population attributable risk may be underestimated, as UK Biobank has a limited age range (mean age of 64 years at first prescription, max 79) that does not include many older people at high risk of strokes or MIs, so estimates of absolute risk may be greater in the general population. Further work is required to model this in other countries. A further limitation is that data from patients treated soon after a stroke are limited by the absence of information on hospital prescribing and the inclusion of patients who received two or more consecutive clopidogrel prescriptions in the community, consistent with our focus on longer-term outcomes in the community. The prescribing data from primary care is designed to be comprehensive for community-based subjects. The only other route to receipt of clopidogrel is for subjects who had periods as hospital inpatients, but in the UK system care is routinely handed back to primary care on hospital discharge, with very short term (usually up to 7 days) medication prescribed. The availability of only primary care prescribing (with no hospital prescribing details available) may mean that some clopidogrel prescriptions have been missed and therefore some patients or exposure periods excluded, but likely only for periods during and soon after hospital admissions. In our primary analysis we excluded events within 7 days of first known prescription, to give clopidogrel time to have a preventative effect: in sensitivity analysis we included these events as patients may have received a prescription before their first in primary care, and we found that results were highly consistent, although with modestly altered estimates. Although this is a relatively large study of 7483 patients, stratified analyses (such as *2 heterozygous/homozygous groups, or by aspirin coprescribing) were underpowered. The analysis was restricted to UK Biobank participants of genetically European ancestry (93% of the cohort); only 492 non-European participants had sufficient clopidogrel data, but this included all other ancestries including South or East Asian, African and mixed, so analysis was not possible. Future analyses with more data will be needed to address these research questions. Similarly, analyses of the low frequency variants in larger samples may be needed to refine estimates, including for rs187941554, for which our nominally significant association (HR 5.44, 95% CIs 1.33 to 22.30, p=0.019) was based on only two strokes in 32 subjects.

To conclude, intermediate and low metabolisers (due to *CYP2C19* LOF alleles) who were prescribed clopidogrel in the primary care setting had substantially increased risks of incident hospital-diagnosed strokes and MIs, compared with normal metabolisers. Evidence for improved patient outcomes with *CYP2C19* genotype-guided antiplatelet therapy is increasing, especially in the acute setting. Genotype-guided section of antiplatelet medications may improve outcomes for primary care patients carrying *CYP2C19* variants.

**Author affiliations**
[1]Epidemiology and Public Health group, College of Medicine and Health, University of Exeter, Exeter, UK
[2]Connecticut Convergence Institute for Translation in Regenerative Engineering, University of Connecticut Health, Farmington, Connecticut, USA
[3]University of Connecticut Center on Aging, University of Connecticut Health, Farmington, Connecticut, USA
[4]National Institute on Aging NIA-ASTRA Unit, Harbor Hospital, Baltimore, Maryland, USA
[5]Exeter Diabetes Group (ExCEED), College of Medicine and Health, University of Exeter, Exeter, UK

**Acknowledgements** Access to UK Biobank resource was granted under application number 14631. We would like to thank UK Biobank participants and coordinators for this dataset. The authors would like to acknowledge the use of the University of Exeter High-Performance Computing (HPC) facility in carrying out this work.

**Contributors** LCP performed the analysis, interpreted results, created the figures, searched literature and cowrote the manuscript. DT and HF performed the literature search, performed analyses and contributed to the manuscript. JLA and JD performed analyses and contributed to the manuscript. C-LK, GAK, LF and JB contributed to data interpretation and contributed to the manuscript. JAHM provided expert clinical interpretation of the data and contributed to the manuscript. DM oversaw data analysis, interpretation and literature searching, and led the writing of the manuscript. DM is the guarantor. The corresponding author attests that all listed authors meet authorship criteria and that no others meeting the criteria have been omitted.

**Funding** LCP and DM are supported by the University of Exeter Medical School and the University of Connecticut School of Medicine. DT is funded by the Ministry of National Education, Republic of Turkey. JLA was supported by an award to DM by the UK Medical Research Council (MR/S009892/1). JD is also supported by the Alzheimer's Society (grant: 338 (AS-JF-16b-007)). GAK is supported by the Travelers Chair in Geriatrics and Gerontology. JB is funded by an Expanding Excellence in England (E3) research grant awarded to the University of Exeter. LF is supported by the National Institute on Aging, NIH. JAHM is funded by a National Institute for Health Research Fellowship (NIHR301445).

**Disclaimer** This publication presents independent research funded by the National Institute for Health Research (NIHR). The views expressed are those of the author(s) and not necessarily those of the NHS, the NIHR or the Department of Health and Social Care. The funders had no input in the study design; in the collection, analysis, and interpretation of data; in the writing of the report; or in the decision to submit the article for publication. The researchers acted independently from the study sponsors in all aspects of this study.

**Competing interests** All authors have completed the ICMJE uniform disclosure form (at www.icmje.org/coi_disclosure.pdf) and declare: no support from any organisation for the submitted work; no financial relationships with any organisations that might have an interest in the submitted work in the previous three years; no other relationships or activities that could appear to have influenced the submitted work.

**Patient consent for publication** Not applicable.

**Ethics approval**  This study involves human participants and was approved by The North West Multi-Centre Research Ethics Committee approved the collection and use of UK Biobank data (Research Ethics Committee reference 11/NW/0382). Participants gave informed consent to participate in the study before taking part.

**Provenance and peer review**  Not commissioned; externally peer reviewed.

**Data availability statement**  Data may be obtained from a third party and are not publicly available. Data are available on application to the UK Biobank ( www.ukbiobank.ac.uk/register-apply). Additional data are available from the corresponding author on reasonable request.

**ORCID iDs**
Luke C Pilling http://orcid.org/0000-0002-3332-8454
Deniz Türkmen http://orcid.org/0000-0002-9584-5625
Janice L Atkins http://orcid.org/0000-0003-4919-9068
Joao Delgado http://orcid.org/0000-0003-1648-871X

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
