## [Reviewer comments · BMJ Open]

ARTICLE DETAILS

TITLE (PROVISIONAL)	Analysis of CYP2C19 genetic variants with ischaemic events in UK patients prescribed clopidogrel in primary care: a retrospective cohort study
AUTHORS	Pilling, Luke; Türkmen, Deniz; Fullalove, Hannah; Atkins, Janice; Delgado, Joao; Kuo, Chia-Ling; Kuchel, George; Ferrucci, Luigi; Bowden, Jack; Masoli, Jane; Melzer, David

VERSION 1 – REVIEW

REVIEWER	Narasimhalu, Kaavya National Neuroscience Institute - Singapore General Hospital Campus
REVIEW RETURNED	16-Jul-2021

GENERAL COMMENTS	This study is a retrospective analyses involving participants of the UK Biobank project who were prescribed clopidogrel. The main findings are that patients with Cyp2C19 LOF mutations were at higher risk of stroke and MI compared to patients with WT CYP2C19. The methods were appropriate for this sample. The manuscript can be improved by amending the following: 1. The abstract focuses more on the ischemic stroke outcomes rather than giving a balanced review (stroke, MI, combined outcomes).2. Inclusion of bleeding events in analyses focusing on the 17* subtype3. It is not clear if the prescription data is comprehensive? Does this cover all national prescriptions? Is there info on sens/spef of these prescription registries available? Biases to do with more than half of the Blobank participants being excluded due to lack of prescription information need to be discussed.4. The large majority of patients appear to have been prescribed clopidogrel after an MI or after a diagnosis of HTN. Roughly 10% appear to be after a stroke diagnosis. Based on the definition of having 2 consecutive prescriptions of clopidogrel, patients who received DAPT post stroke based on CHANCE, POINT, THALES trials are not likely to have been captured in this study (though the impact overall will be low given that only 10% are presumptively post stroke).
---

REVIEWER	Claassens, Daniel Saint Antonius Hospital
REVIEW RETURNED	18-Jul-2021

GENERAL COMMENTS	This analysis by Pilling et al. investigates whether CYP2C19 loss-of-
---

	function variants increase the risk of ischemic Major comments: The authors mention that they exclude events within the first 7 days after clopidogrel prescription to prevent including events before the full preventative effect of clopidogrel is seen. Therefore they exclude 5-7% of stroke and myocardial infarction events. However, they also mention they do not have access to hospital data. Most patients using clopidogrel will use this due to an acute coronary syndrome or ischemic stroke, which is initiated during hospital admission. In most cases, patients will therefore have already used clopidogrel prior to first outpatient prescription. Are results different when including these events? Moreover, is it necessary to exclude these events? The manuscript could be shortened by removing sections concerning other genetic variants which are very rare (<3% for CES 1 and <0.3% for the others), meaning any significant results have a high likelihood of being impacted by chance, and in addition their clinical relevance is questionable. Minor comments: I would recommend simplifying the objective in the abstract to make it easier to read (the section between brackets). In the strengths and limitation section, I do not understand what the authors mean with “or up to 18 years”. Perhaps this could be rewritten or clarified. When referring to the outcomes measures, the authors often include the term incident throughout the manuscript, like in “incident hospital-diagnosed ischaemic stroke”. However, I feel that this makes the text unnecessarily harder to understand for the average reader and wonder whether it is necessary to include the term. Authors use references 25-27 to refer to studies investigating alternative treatment strategies in CYP2C19 LoF allele carriers. However, these are only very small studies (a few 100 patients) with a limited number of patients. Instead, it would be better to refer to the larger trials like POPular Genetics (Claassens et al, NEJM 2019) or TAILOR PCI (Pereira et al, JAMA 2020) (Both 1000+ patients).
--	--

VERSION 1 – AUTHOR RESPONSE

Reviewer: 1

Dr. Kaavya Narasimhalu, Natl Univ Singapore

Comments to the Author:

This study is a retrospective analyses involving participants of the UK Biobank project who were prescribed clopidogrel. The main findings are that patients with Cyp2C19 LOF mutations were at higher risk of stroke and MI compared to patients with WT CYP2C19. The methods were appropriate for this sample.

The manuscript can be improved by amending the following:

1. The abstract focuses more on the ischemic stroke outcomes rather than giving a balanced review (stroke, MI, combined outcomes).

We have modified the abstract to better emphasise the MI and combined result. The “Results” section now reads:

28.7% of participants carried at least one CYP2C19 LoF variant. LoF carriers had higher rates of incident ischaemic stroke whilst treated with clopidogrel compared to those without the

variants (8 per 1,000 person-years versus 5.2 per 1,000 person-years; Hazard Ratio 1.53: 95% Confidence Intervals 1.04 to 2.26, $p=0.031$). **LoF carriers also had increased risk of MI (HR 1.14: 95%CI 1.04 to 1.26, $p=0.008$). In combined analysis LoF carriers had increased risk of any ischaemic event (stroke or MI) (HR 1.17: 95%CI 1.06 to 1.29, $p=0.002$).** Adjustment for aspirin co-prescription produced similar estimates. In lifetables using observed incidence rates, 22.5% (95%CI 14.4% to 34.0%) of CYP2C19 LoF carriers on clopidogrel were projected to develop an ischaemic stroke by age 79 (oldest age in the study), compared to 15.4% (95%CI 11.4% to 20.5%) in non-carriers, i.e. 7.1% excess stroke incidence in LoF carriers by age 79.

2. Inclusion of bleeding events in analyses focusing on the 17* subtype

Using ICD-10 codes for Major Bleeding from a recent paper (DOI [10.1111/jep.13400](https://doi.org/10.1111/jep.13400)) we performed this analysis for CYP2C19 *17 carriers and did not find a significant association. We have included the following text in the manuscript:

Methods – “Disease ascertainment” section

...

In sensitivity analysis we investigated major bleeding using ICD-10 codes from (21) (see Supplementary Table 3 for codes used).

Results – end of “CYP2C19 LoF associations with incident ischaemic stroke” section with *17 analysis

...

In analysis of Major Bleeding whilst prescribed clopidogrel CYP2C19 *17 carriers were not at significantly greater or lesser risk compared to non-carriers (n events during prescribing period=167; OR 0.88: 95% CI 0.62 to 1.24, $p=0.46$).

In the Supplementary Methods table we note that we excluded three of the ICD-10 codes (N95.0, K62.5 and R31.* i.e. "Unspecified haematuria" or "Haemorrhage of anus and rectum" or "Post-menopausal bleeding") because there were >10,000 UK Biobank participants with each code, which in our view was too common to be “Major” bleeding for most participants. Results were unaffected either way.

3. It is not clear if the prescription data is comprehensive? Does this cover all national prescriptions? Is there info on sens/spec of these prescription registries available? Biases to do with more than half of the Biobank participants being excluded due to lack of prescription information need to be discussed.

Thank you for the comment. We have added the following to the discussion to clarify the data source:

... The prescribing data from primary care is designed to be comprehensive for community-based subjects. The only other route to receipt of clopidogrel is for subjects who had periods as hospital inpatients, but in the UK system care is routinely handed back to primary care on hospital discharge, with very short term (usually up to 7 days) medication prescribed. The availability of only primary care prescribing (with no hospital prescribing details available) may mean that some clopidogrel prescriptions have been missed and therefore some patients or exposure periods excluded, but likely only for periods during and soon after hospital admissions. In our primary analysis we excluded events within 7 days of first known prescription, to give clopidogrel time to have a preventative effect: in sensitivity analysis we included these events as patients may have received a prescription before their first in primary care, and we found that results were highly consistent, albeit with modestly altered estimates.

4. The large majority of patients appear to have been prescribed clopidogrel after an MI or after a diagnosis of HTN. Roughly 10% appear to be after a stroke diagnosis. Based on the definition of having 2 consecutive prescriptions of clopidogrel, patients who received DAPT post stroke based on

CHANCE, POINT, THALES trials are not likely to have been captured in this study (though the impact overall will be low given that only 10% are presumptively post stroke).

Thank you for raising. Whilst we agree with the point, due to our focus on longer-term outcomes in this study we believe it is appropriate to include these criteria. We have added the following to the discussion:

A further limitation is that data from patients treated soon after a stroke are limited by the absence of information on hospital prescribing and the inclusion of patients who received 2 or more consecutive clopidogrel prescriptions in the community, consistent with our focus on longer term outcomes in the community.

Reviewer: 2

Dr. Daniel Claassens, Saint Antonius Hospital

Comments to the Author:

This analysis by Pilling et al. investigates whether CYP2C19 loss-of-function variants increase the risk of ischemic

Major comments:

The authors mention that they exclude events within the first 7 days after clopidogrel prescription to prevent including events before the full preventative effect of clopidogrel is seen. Therefore they exclude 5-7% of stroke and myocardial infarction events. However, they also mention they do not have access to hospital data. Most patients using clopidogrel will use this due to an acute coronary syndrome or ischemic stroke, which is initiated during hospital admission. In most cases, patients will therefore have already used clopidogrel prior to first outpatient prescription. Are results different when including these events? Moreover, is it necessary to exclude these events?

We took the decision to exclude as this seemed the safest option to avoid potential bias including events that clopidogrel could not possibly have prevented. However, it is a fair point (and limitation we discuss) that we only have prescriptions made in the primary care setting and therefore for some patients our “date of first prescription” is not the first prescription they ever receive. We take the point that we may be unnecessarily excluding this period and have repeated the analysis without the exclusion criteria (i.e. in secondary analysis we included all events, even those within 7 days of first known prescription. For MIs, this adds 126 events (1,823 to 1,949 total) and the associations remains highly consistent – the estimates change from (HR 1.14: 95% CI 1.04 to 1.26, p=0.008) to (HR 1.16: 95% CI 1.06 to 1.28, p=0.021). For strokes this adds 6 events (110 to 116 total) – that in our view includes strokes that may not be valid - and changes the results from (HR 1.53: 95% CI 1.04 to 2.26, p=0.031) to (HR 1.45: 95% CI 0.99 to 2.13, p=0.053). Though the 95% confidence intervals cross the null, the association is highly consistent.

Additionally, there is extensive data on outcomes of clopidogrel prescribing by genotype in the acute treatment settings, and even in the first year of treatment. We therefore focussed the current study on longer term outcome of treatment in the community, an area where little data has been available. In our sample, 33% of those prescribed clopidogrel had not had a previous MI or stroke. Our sample includes both those started on clopidogrel in hospitals, as well as those likely started in the community as part of cardiovascular prevention.

We have modified the manuscript to include the following sentences:

Methods – Statistical analysis section

...

Patients entered the model 7 days after the date of first prescription of clopidogrel; this was to exclude the small number of events that occurred immediately after being prescribed clopidogrel, before it could reasonably have had a preventative effect. Seven days was chosen because the effect of clopidogrel on platelet aggregation was significantly different

between CYP2C19 genotype groups after 7-10 days in a study of 375 patients (22). **In sensitivity analysis we included all events during the prescribing period.** Patients exited the model on the last known date of clopidogrel prescription.

...

Results – Ischaemic stroke section

...

We also performed analysis including all events during the prescribing period (the primary analysis excluded events in the first 7 days of prescribing because the effect of clopidogrel on platelet aggregation was significantly different between CYP2C19 genotype groups after 7-10 days in a study of 375 patients (22)). This period included 6 strokes and changed the estimates from (HR 1.53: 95% CI 1.04 to 2.26, p=0.031) to (HR 1.45: 95% CI 0.99 to 2.13, p=0.053). Though the 95% CIs cross the null the association is highly consistent, despite inclusion of some possibly invalid events.

...

Results – MI section

...

Sensitivity analysis including all MIs during the prescribing period (i.e. not excluding those within 7 days of first known prescription) included 126 more events and changed the estimates from (HR 1.14: 95% CI 1.04 to 1.26, p=0.008) to (HR 1.16: 95% CI 1.06 to 1.28, p=0.021).

...

Discussion – to limitations section

...

The availability of only primary care prescribing (with no hospital prescribing details available) may mean that some clopidogrel prescriptions have been missed and therefore some patients or exposure periods excluded, but likely only for periods during and soon after hospital admissions. **In our primary analysis we excluded events within 7 days of first known prescription, to give clopidogrel time to have a preventative effect: in sensitivity analysis we included these events as patients may have received a prescription before their first in primary care, and we found that results were highly consistent, albeit with modestly altered estimates.**

...

The manuscript could be shortened by removing sections concerning other genetic variants which are very rare (<3% for CES 1 and <0.3% for the others), meaning any significant results have a high likelihood of being impacted by chance, and in addition their clinical relevance is questionable.

With respect we do not see the value in removing this (short) section. We tested the hypothesis that reported variants affecting clopidogrel were associated with strokes, and found a negative result (this could be due to power, or indeed a true negative result).

Minor comments:

I would recommend simplifying the objective in the abstract to make it easier to read (the section between brackets).

We have removed the text in the brackets from abstract line 1.

In the strengths and limitation section, I do not understand what the authors mean with “or up to 18 years”. Perhaps this could be rewritten or clarified.

We have completely redrafted this section, which now reads:

- A strength of this study is that it is a relatively large study of 7,483 adults prescribed clopidogrel for up to 18 years in the primary care setting, with follow-up in electronic medical records.
- A limitation of the study is that it is based on retrospective cohort data from the UK Biobank, which is not population representative.
- A strength of the study is that linked electronic medical records (for prescribing and outcome data) is unaffected by sample response patterns and is thus more representative.
- A limitation of the study is that we only had access to prescriptions made in the primary care setting, and we therefore miss any prescriptions made in the hospital setting.
- A limitation of the study is the limited number of older people included (max age reached was 79 years) where the incidence of thrombotic events is highest, thus the absolute incidence rates of ischaemic events in the whole cohort were low.

When referring to the outcomes measures, the authors often include the term incident throughout the manuscript, like in “incident hospital-diagnosed ischaemic stroke”. However, I feel that this makes the text unnecessarily harder to understand for the average reader and wonder whether it is necessary to include the term.

We were referring the incident events that occur after initiation of clopidogrel treatment. We have simplified the text to use the word “incident” fewer times (e.g. start of section, and from then on just say “strokes”).

Authors use references 25-27 to refer to studies investigating alternative treatment strategies in CYP2C19 LoF allele carriers. However, these are only very small studies (a few 100 patients) with a limited number of patients. Instead, it would be better to refer to the larger trials like POPular Genetics (Claassens et al, NEJM 2019) or TAILOR PCI (Pereira et al, JAMA 2020) (Both 1000+ patients).

Our thanks for highlighting these studies, and apologies for missing them. We have included them in the Discussion and redrafted the text as follows:

Other recent studies in different populations, especially in patients with acute coronary syndrome **and those undergoing primary percutaneous coronary intervention such as the large POPular Genetics trial**, also support this conclusion (25–27,**34,35**).

VERSION 2 – REVIEW

REVIEWER	Narasimhalu, Kaavya National Neuroscience Institute - Singapore General Hospital Campus
REVIEW RETURNED	30-Sep-2021
GENERAL COMMENTS	The authors have addressed my queries.
REVIEWER	Claassens, Daniel Saint Antonius Hospital
REVIEW RETURNED	14-Oct-2021
GENERAL COMMENTS	The authors have adequately addressed all my previous comments and I have no further comments.